# Effect of Serum and Oxygen on the In Vitro Culture of Hanwoo Korean Native Cattle-Derived Skeletal Myogenic Cells Used in Cellular Agriculture

**DOI:** 10.3390/foods12071384

**Published:** 2023-03-24

**Authors:** Sun A Ock, Kang-Min Seo, Won Seok Ju, Young-Im Kim, Ha-Yeon Wi, Poongyeon Lee

**Affiliations:** Animal Biotechnology Division, National Institute of Animal Science (NIAS), Rural Development Administration (RDA), 1500, Kongjwipatjwi-ro, Isero-myeon, Wanju-gun 565-851, Republic of Korea

**Keywords:** high serum levels, hypoxia, culture medium, Hanwoo, cellular agriculture

## Abstract

Skeletal muscle-derived myogenic cells (SKMCs) are novel protein sources capable of replacing animal meat. However, SKMCs have not been commercialized owing to poor productivity and the high cost of in vitro cell culture. Therefore, we cultured SKMCs in varying serum (5–20%) and oxygen concentrations (5–20%) to investigate the parameters that most impact cell productivity (serum, hypoxia, and culture medium) and examined cell proliferation ability and genes involved in myogenesis/proliferation/apoptosis/reactive oxygen species (ROS). In fetal bovine serum (FBS) groups, hypoxia induction doubled cell number, and the 20% FBS/normoxia group exhibited similar cell numbers as 5% FBS/5% hypoxia, confirming that 5% hypoxia reduced serum requirement by four-fold. The use of 20% FBS downregulated *MTF5*/*MYOD1*/*MYOG*/*MYH1*, whereas hypoxia induction with ≤10% FBS upregulated them. Although 20% FBS lowered *TERT* expression through rapid cell proliferation, *NOX1*, a major factor of ROS, was suppressed. DMEM/F12 demonstrated better differentiation potential than F10 by upregulating *MYF3*/*MYOD1*/*MYOG*/*MYH1* and downregulating *MSTN*, particularly DMEM/F12 with 2% FBS/5% hypoxia. The myogenic fusion index was higher in DMEM/F12 without FBS than in DMEM/F12 with FBS (0.5–5%); however, the total nuclei number was reduced owing to apoptosis. Therefore, high serum levels are essential in influencing SKMC growth, followed by hypoxia as a synergistic component.

## 1. Introduction

Studies on in vitro proliferation and differentiation of skeletal muscle cells have recently expanded beyond medical applications to include techniques for boosting cattle output. In particular, the idea of cellular agriculture is being explored as an alternative solution for the supply of animal protein in extreme environments experiencing an imbalance in the supply and demand of animal protein, along with the development of animal biotechnology [1,2].

In contrast to typical culture conditions, in vitro growth medium supplemented with greater than or equal to 20% serum is required for myoblast proliferation [1,2,3]. Therefore, reducing or replacing the amount of serum added to the culture medium without lowering cellular agriculture productivity, in terms of safe and steady productivity increase, animal welfare, and avoidance of deadly animal infectious diseases, is critical. Many researchers and commercial culture medium developers have sought to develop chemically defined cultures that substitute albumin, fetuin, hormones, vitamins, and growth factors present in serum; accordingly, some groups have developed B8 and B9 media [2,4]. However, in terms of efficacy, these culture media are ineffective for long-term culture and can only attain price competitiveness when a mass media production system is established [1,5,6].

In vitro cell culture is performed in an incubator with ~20% oxygen (atmospheric oxygen, normoxia) and 5% CO_2_, exposing the cells to the risk of delayed proliferation and aging resulting from exposure to higher oxygen levels than the body’s oxygen concentration [7]. The oxygen concentration of skeletal muscle in the body, which contains skeletal muscle stem cells, is reportedly between 1 and 10% (physical oxygenation and physiological hypoxia) [8,9]. Therefore, some studies sought to culture myocytes under various hypoxic conditions to induce high cell proliferation in vitro [9,10,11]. However, previous research on hypoxia and normoxia during myocyte development has reported inconsistent results [12]. Some studies reported that normoxia is more efficient than hypoxia for in vitro differentiation of myocytes, whereas others reported that hypoxia is more efficient than normoxia for myocyte in vitro differentiation [7,13]. Most publications suggest that normoxia has a stronger differentiating capacity than physiological hypoxia, with results obtained using an immortalized mouse myoblast cell line C2C12 [10,13]. Therefore, accurate additional analysis that reflects the characteristics of normal muscle cells and source species and the composition of the culture medium is required.

Previous studies have separately analyzed the factors that affect skeletal muscle cell growth and differentiation. However, this approach has made it unclear which factors are most important. In this study, we investigated the correlation between serum concentration and physiological hypoxia as important variables influencing the in vitro proliferation of skeletal muscle cells derived from Korean native cattle (Hanwoo) and the formation of multinucleated myotubes. Additionally, we investigated the primary influence of differentiation induction medium composition, such as oxygen level, in a controlled environment to elucidate previously contradictory results. Unlike previous studies, this study is unique in using a serum-free differentiation medium to successfully induce multinucleated myotubes from slaughterhouse-derived fresh beef skeletal muscle myocytes.

## 2. Materials and Methods

### 2.1. Reagents and Media

Unless otherwise specified, all chemicals were purchased from Sigma-Aldrich Corporation (St. Louis, MO, USA), and the media were obtained from Thermo Fisher Scientific (San Diego, CA, USA).

### 2.2. Isolation and Culture of Skeletal Myogenic Cells (SKMCs) from Bovine Skeletal Muscle Tissue

Immediately following the killing of three Korean native bulls (Hanwoo, 2 years old), skeletal muscle tissues were obtained from the semitendinosus, stored on ice, and moved to the laboratory within an hour. Each skeletal muscle tissue (~100 g) was washed several times with phosphate-buffered saline (PBS), followed by the addition of 1× antibiotic-antimycotic solution. The tendon, fat tissue, and exterior muscle tissue were removed to avoid microbial contamination. The skeletal muscle tissue of ~1 cm in width and height was finely chopped with a blade, treated with 0.25% collagenase (STEMCELL technology, Vancouver, BC, Canada), and incubated for 30–60 min at 37 °C (Figure 1). The muscle fragments and cells were treated with advanced Dulbecco’s modified Eagle’s medium (ADMEM) containing 10% fetal bovine serum (FBS) and 1× antibiotic-antimycotic solution and then centrifuged at 1000× *g* for 5 min. To extract mononuclear myogenic cells, the pellet was resuspended in ADMEM with 10% FBS and 1× antibiotic-antimycotic solution and filtered with 100- and 40-μm cell strainers. To recover SKMC-enriched cells, the cells were cultured using either the conventional pre-plating isolation method [14,15] or the slowly adherent cell culture method based on the difference in adhesion speed across cell types (Figure 1A).

An adaptation of a method for eliminating cells with rapid adhesion ability is depicted in Figure 1. Mononuclear myogenic cells were seeded on collagen-coated T-75 flasks (Nunc EasYFlask 75 cm^2^, Thermo Fisher Scientific) and cultured for 2 h. The supernatant containing non-adhesion cells was transferred to a new collagen-coated T-75 flask and cultured in an incubator with 5% CO_2_ at 37 °C until the cells reached ~60% confluence. Every three days, the growth medium (GM) was replaced with new culture media. The cells were harvested with 0.25% trypsin-EDTA and washed twice using ADMEM with 10% FBS and 1× penicillin–streptomycin. This procedure was repeated, and groups 1 and 2 were treated differently from this point forth: for Group 1, this procedure was performed once more. For Group 2, the cells were treated with the antibodies human allophycocyanin (hAPC)-CD29^+^ (BioLegend Inc., San Diego, CA, USA)/hAPC-CD59^+^ (BD Biosciences, San Jose, CA, USA), and they were detected using anti-APC microbeads (Miltenyi Biotec, Bergisch Gladbach, NRW, Germany). Finally, they were subjected to magnetic-activated cell sorting positive selection using columns [3] (Figure 1B). All cells from both groups were further cultured and cryopreserved for subsequent experiments.

Ear tissue was obtained aseptically from one of the bulls involved in the experiment. The hairs were physically removed, and ear fibroblasts were cultured in vitro using only the dermal tissue [16]. The cells obtained after at least 10 passages were used as a control for genetic analysis.

### 2.3. In Vitro Culture Environment for SKMCs

GM used for the proliferation of SKMCs was divided into two types: PromoCell GM (PromoCell GmbH, Heidelberg, Germany), which comprised skeletal muscle cell basal medium (500 mL) and supplement mix (5% fetal calf serum, 50 μg/mL fetuin, 10 ng/mL epidermal growth factor [EGF], 1 ng/mL basic fibroblast growth factor [bFGF], 10 μg/mL insulin, and 0.4 μg/mL dexamethasone [DEX]) and DMEM/F12 (Gibco 10565-018; Thermo Fisher Scientific)-based GM (DMEM/F12 GM) supplemented with 10 ng/mL bFGF, 10 ng/mL hepatocyte growth factor (HGF), 10 ng/mL insulin-like growth factor 1 (IGF1), 10 ng/mL EGF, 2 ng/mL transforming growth factor beta 1 (TGFβ1), 10 nM DEX, and 1× penicillin–streptomycin. The concentration of FBS, added for the proliferation of SKMCs, was adjusted to 5–20% for the experiment. The washing medium for SKMCs was ADMEM supplemented with 10% FBS and 1× penicillin–streptomycin.

Differentiation medium (DM) for SKMCs was divided into three types: PromoCell DM (serum-free medium), which comprised skeletal muscle cell DM supplemented with 10 μg/mL insulin, F10 DM, and DMEM/F12 DM with 10 ng/mL IGF1. Concentrations of FBS added for the differentiation of SKMCs were adjusted to 0–5% (0, 0.5, 2, and 5%) depending on the purpose of the experiment. GM or DM was replaced every three days with a fresh culture medium. The incubator environment for SKMCs was set at 37 °C, 100% humidity, and 5% CO_2_. Additionally, the concentration of oxygen in the incubator was adjusted to 5–20%, depending on the purpose of the experiment.

### 2.4. Evaluation of In Vitro SKMC Proliferation

SKMCs were seeded into 24-well plates at a concentration of 1000 cells per well and cultured under six different culture conditions for 14 days as follows: serum at three different concentrations (5, 10, or 20% FBS) in the GM and two different oxygen concentrations (normoxia [20%] or hypoxia [5%]) in the incubator. Every three days, the culture medium was replaced with fresh GM. Every two days, the cells were harvested for cell counting. The number of cells was measured three times per sample using a disposable hemocytometer, and the average value was noted. This experiment was performed with three independent wells.

### 2.5. Quantitative Real-Time PCR

To minimize individual variability, samples for real-time PCR analysis were pooled from three independent cell culture dishes. A previously reported approach was used to extract total RNA, synthesize cDNA, and perform quantitative real-time PCR for specified genes [16]. The specific primer sets used are listed in Appendix A. TATA-box binding protein gene was used as the internal control, and all samples were analyzed five times. Data were analyzed via the 2^−ΔΔCT^ method.

### 2.6. Immunocytofluorescence Staining

The approach described by Ullah et al. [17] was used for immunocytofluorescence staining, pretreatment, counterstaining, and mounting. Following a previous report [3], the primary antibodies used for the detection of specific proteins were those against mouse PAX7 (1:10; Developmental Studies Hybridoma Bank, Iowa, IA, USA) and mouse MYOD (1:200, ABclonal, Cat. No. A0671; Woburn, MA, USA), followed by secondary antibody treatment with Alexa Fluor 488-(1:200; green; Thermo Fisher Scientific) or Alexa Fluor 594 (red)-labeled anti-mouse antibodies.

### 2.7. Myotube Fusion Index (MFI) Analysis

SKMCs of ~90% confluence cultured with GM on collagen type 1-coated 35-mm dish (Thermo Fisher Scientific) were washed twice with DM and cultured with DM for 5–7 days. SKMCs were then washed with Dulbecco’s PBS and fixed with 3.7% formalin (Sigma-Aldrich). The cells were treated overnight at 4 °C with the primary antibody, anti-myosin (Skeletal, Fast) antibody (dilution, 1:400; Sigma-Aldrich). The next day, after washing the cells, they were treated with goat anti-mouse IgG (H+L)-Alexa Fluor 488 (dilution, 1:200; Invitrogen, Waltham, MA, USA) or -Alexa Fluor 594 (dilution, 1:400; Invitrogen) [3] for 1 h. The nuclei were counterstained with 4′,6-diamidino-2-phenylindole (DAPI), or propidium iodide (PI) for 30 min. The number of nuclei in MYH1-positive myotubes was counted and expressed as a percentage of the total. Finally, the samples were observed, and the number of nuclei was counted under a fluorescence microscope (Leica Microsystems, Inc., Wetzlar, Hesse, Germany) and analyzed via a previously modified calculation method [18]. The MFI was calculated by dividing the number of nuclei within the MYH1-positive myotubes (with three or more nuclei) by the total number of nuclei × 100 [3,18]. Two independent collaborators performed the counting via a blinded analysis approach across all the treatment groups.

### 2.8. Statistical Analysis

For statistical analysis, data from at least three replicates were analyzed using one- or two-way analysis of variance (ANOVA), followed by posthoc Tukey or least significant difference (LSD) test using IBM SPSS statistics software, version 25 (SPSS Inc., Chicago, IL, USA). PCR data are expressed as relative quantification (RQ) data, and other data are represented as mean ± standard error of mean (SEM). Statistical significance was set at *p*-value < 0.05.

## 3. Results

### 3.1. Cell Morphology Based on the GM Used

In the first 2 h, many undigested muscle tissues and connective tissue networks were observed, and when the supernatant was removed and cultured, numerous fibroblast-like cells were observed in the tissue culture flasks. When the cells reached the final pre-plating stage, compact and spherical cells with a diameter of ~10 μm that resembled myoblasts or satellite cells were observed but in extremely small numbers. Cells were cultured in two different GMs: commercial PromoCell GM (Control) and DMEM/F12 GM. Cells cultured in PromoCell GM for 1 week revealed reduced cell proliferation and the presence of undesirable cell morphologies, such as senescent cells. However, culturing with DMEM/F12 GM with 20% FBS induced a high proliferation rate and healthy cell morphology (Figure 2). Therefore, DMEM/F12 GM was selected for cell proliferation. However, the pre-plating method did not recover an adequate proportion of cells participating in myogenesis for further cell agriculture. Subsequently, a method for easily and efficiently separating a large number of myogenic cells was tested.

### 3.2. Expression of Genes Encoding Myogenic Regulatory Factors (MRFs)

Figure 1 shows the removal of rapidly adhering cells. The supernatant containing slowly adhering cells was transferred to fresh flasks, and the cells were cultured until they reached 70% confluence. This process was performed iteratively. The cells were extracted via two methods: SAC culture (Group 1) and positive selection of CD29^+^/CD56^+^ cells (Group 2). Figure 3 shows how transcription factor encoding genes involved in the myogenesis pathway are used to assess cell proliferation and differentiation capacity.

*PAX7/MYF5/MYOD1/MYOG* expression levels increased in all groups throughout the proliferation stage, particularly *MYOG* expression in Group 2*. MYF6* expression was reduced in both groups, compared with that in bovine ear fibroblasts.

After seven days of culturing with DM, myotubes developed normally in both groups, with no differences observed between the two groups. *MYF5/MYOD1/MYOG* expression levels were upregulated in cells cultured in DM than in cells cultured in GM in all groups, notably in Group 1. Therefore, cells isolated using the Group 1 approach were used for further experimentation.

### 3.3. Effects of Serum and Oxygen Levels on the Proliferation of SKMCs

The interaction between oxygen and serum levels was analyzed using two-way ANOVA (Figure 4, Appendix A) and confirmed from 4 to 10 days. After ten days, the interaction effect of oxygen and serum levels could not be directly compared because groups with 20% FBS/oxygen had many cells that spontaneously fell out from the bottom of the well. When the SKMCs were cultured in 5, 10, or 20% FBS, the group exposed to hypoxia had a higher number of cells than the group exposed to normoxia. In groups subjected to 5% hypoxia, increasing FBS concentration resulted in an increase in cell number.

### 3.4. Effects of PAX7/MRF Expression on the Proliferation of SKMCs

As shown in Figure 5 and Appendix A, SKMCs were cultured in different FBS and oxygen concentrations for 14 days, and expression levels of myogenesis-related transcription factors were assessed on day 7 (Figure 5A) and 14 (Figure 5B). The interaction effect between FBS and oxygen levels was analyzed using the Δ*CT* of each gene, and it was as high as ≥0.5 on days 7 and 14 based on the η^2^*p* value, particularly on day 14 (Appendix A). When η^2^ values of all genes were analyzed, their expression exhibited a greater correlation with FBS concentrations than with oxygen concentrations, independent of culture duration. Therefore, it was confirmed that FBS and oxygen concentrations have a close impact on the expression of genes involved in cell proliferation, with FBS having a greater influence on gene expression than oxygen.

On day 7 of proliferation, groups with 20% FBS exhibited a greater upregulation of *MYF5/MYOD1/MYOG/MYH1* than those with ≤10% FBS. *MYF5/MYOD1/MYOG/MYH1* was upregulated by hypoxia induction in groups with 10% FBS (Figure 5). The groups with ≤10% FBS exhibited comparable expression patterns of the genes analyzed on days 7 and 14 of growth. In the group with ≤10% FBS, the induction of hypoxia upregulated *MYF5/MYOD1/MYOG/MYH1*, resulting in a similar expression pattern of these genes to that observed on day 7.

### 3.5. Effect of Serum and Oxygen Levels on Cell Lifespan-Related Genes during In Vitro SKMC Growth

Genes involved in proliferation and apoptosis were analyzed in cells cultured in vitro for 14 days. The effects of FBS and oxygen levels were analyzed, and a close interaction was confirmed between FBS and oxygen levels based on η^2^ and η^2^*p* values, as shown in Figure 6 and Appendix A. FBS levels had a higher correlation with the expression of genes involved in proliferation and apoptosis, except *BCL2*, than oxygen levels based on the η^2^ values (Appendix A).

The reduction in the FBS concentration from 20% to ≤10% upregulated *TERT* (≥~2-fold) and downregulated *CDKN1A*. In the 20% FBS group*, CDKN1A* expression decreased as oxygen levels were decreased. *TP53* was downregulated (~0.5-fold) in groups with 5% FBS than in the ≥10% FBS group. *MYC* was upregulated, or its expression was maintained in 5% more hypoxia than in normoxia. *BAX* was downregulated (~0.5 fold) in the 5% FBS and ≥10% oxygen group. *BCL2* was downregulated in ≤10% hypoxia groups with 20% or 5% FBS but not in those with 10% FBS.

### 3.6. Effect of FBS and Oxygen Levels on Reactive Oxygen Species (ROS) during In Vitro SKMC Growth

As shown in Figure 6B and Appendix A, the interaction effect of FBS and oxygen levels on ROS-related genes was analyzed in cells cultured in vitro for 14 days. Based on η^2^ (≥0.06) and η^2^*p* (≥0.14) values, there was a close interaction between FBS and oxygen levels on the expression of ROS-related genes, except *GPX1*. FBS levels had a higher correlation with the expression pattern of these genes than oxygen levels based on η^2^ value (Appendix A and Appendix A).

NADPH oxidase 2 (*NOX2*) was upregulated in the ≤10% FBS group than in the 20% FBS group, particularly in the 5% FBS and normoxia group. *GPX1* was only downregulated (~0.5-fold) in the 5% FBS groups. Additionally, *SOD* expression, particularly of *SOD3*, decreased as FBS concentration decreased. Furthermore, *SOD* was downregulated by ≤~0.5-fold in the 20% FBS and hypoxia group. Overall, the expression patterns of ROS-related genes were affected more by FBS than by hypoxia.

Finally, as shown in Appendix A, SKMCs were cultured in GM with 20% FBS under hypoxia, and the expression of *HIF*, *p38-MAPK*, *ERK*, and *mTOR*, which react with oxygen in the myogenic pathway, as well as *MSTN*, a myogenic inhibitory gene, were analyzed. *HIF*, *p38-MAPK-14*, *ERK*, and *mTOR* exhibited no significant differences in expression when hypoxia was induced; however, *MSTN* expression had an inverse correlation with oxygen concentration.

### 3.7. SKMC Differentiation Potential in Different DMs

The in vitro differentiation capacity of SKMCs was analyzed with three different media: PromoCell, F10, and DMEM/F12 basal DM. As indicated in Figure 7, the differentiation capacity of SKMCs was preferentially examined with PromoCell without FBS under varying oxygen conditions. On day 7, myotubes were formed (Figure 7C), followed by the analysis of *MYH1*/*MSTN* expression and MFI. *MYH1*/*MSTN* expression had a 1.57-fold difference across groups; however, MFI had no significant difference. Morphologically enlarged flat senescent cells and multinucleated myotubes with enlarged vacuoles were frequently observed in cells cultured in PromoCell DM. Notably, cell death dislodged numerous cells from the dish surface.

In the next step, SKMCs were cultured with F10 (Figure 7A) or DMEM/F12 DM (Figure 7B), and the expression of genes encoding MRFs and *MYH1*/*MSTN* was analyzed (Figure 8). Except for *MYF6*, all genes were upregulated in cells cultured in F10 DM than in those cultured in GM. Among all groups, *MTH1* was most highly upregulated (2.6-fold compared with that in cells cultured in GM) in the 2% FBS/10% hypoxia group. *MSTN* expression was the lowest in the group subjected to 5% FBS/5% hypoxia (3.2-fold compared with that in cells cultured in GM). *MYF5*/*MYOD1*/*MYOG* was upregulated in cells cultured in DMEM/F12 DM than in cells cultured in GM; particularly, *MYF5* expression showed minimum upregulation by 13.9-fold and maximum upregulation by 116.7-fold. *MYH* was also drastically upregulated in cells cultured in DMEM/F12 DM than in cells cultured in GM (16.5–167.1-fold), particularly under 2% FBS/5% hypoxia, and its expression was inversely correlated with oxygen concentration. *MSTN* expression was the lowest in the 5% FBS/10% hypoxia group (1.4-fold compared with that in cells cultured in GM). When F10 DM and DMEM/F12 DM were used to differentiate SKMCs, DMEM/F12 DM demonstrated greater differentiation capacity than F10 DM owing to the substantial increase in *MYH* expression and inhibition of *MSTN* expression.

As indicated in Appendix A, the interaction effect between FBS and oxygen levels on the regulation of myogenic regulators by different DMs was evaluated. The interaction effect between FBS and oxygen levels was confirmed using two-way ANOVA. In F10 DM, η^2^ values of myogenic regulators were higher in expression in response to FBS than in response to oxygen, except for *MYF5* expression, thereby confirming that FBS has a closer correlation with SKMC differentiation. However, in DMEM/F12 DM, the opposite values for FBS and oxygen levels were observed, except for *MYF5*/*MSTN* expression, thereby confirming that oxygen concentration has a close correlation with SKMC differentiation. Therefore, the association between FBS and oxygen levels depended on the composition of DM. Based on the above experimental results, the next test was performed with DMEM/F12 DM.

### 3.8. MFI Based on the FBS Concentration Added to DMEM/F12 DM

As shown in Figure 9, SKMCs were cultured in DM with 0–5% FBS at normoxia condition for 7 days, and the formation of multinucleated myotube was observed (Appendix A) and MFI was evaluated. The overall number of nuclei was reduced in DM groups than in GM groups; however, there was no change based on FBS concentrations (0.5% FBS group, 75.38 ± 3.03; 2% FBS group, 73.14 ± 0.95; and 5% FBS group, 95.86 ± 5.51). The overall number of nuclei in the DM group without FBS (46.76 ± 4.35) was lower than that in other DM groups with FBS (Figure 9A), and many cells in the surrounding myotubes were lost (Figure 9B). The number of nuclei in myotubes increased more in the DM groups than in the GM groups; however, no difference was observed among the DM groups. The number of myotubes was increased in the DM groups (1.86–2.24) than in the GM group (0.71). MFI did not differ among DM groups with FBS and the DM group without FBS (19.62 ± 1.85) and had the highest MFI among the DM groups as per the LSD test based on the lowest total number of nuclei.

## 4. Discussion

This study investigated the impact of three important factors, serum (FBS), hypoxia, and culture medium, on the in vitro culture of skeletal muscle cells recovered from adult cattle following slaughter. Skeletal muscle cells relied more on serum than on hypoxia during in vitro proliferation. Owing to high levels of serum, 20% FBS inhibited *MYOD*/*MYOG*/*MYH1* expression, reduced ROS by inhibiting *NOX2* expression and maintaining *SOD3* expression, and stabilized *HIF* expression. The muscle growth inhibitory gene *MSTN* was greatly affected by microenvironmental factors, such as SKMC growth and differentiation conditions. The key parameters influencing *MSTN* expression inhibition during differentiation were DM composition, FBS, and hypoxia, in that order.

Isolation of pure skeletal muscle satellite cells, muscle stem cells, and myoblasts is required for precise research in muscle regeneration medicine, and most studies have used the following two major isolation methods: sorting using specific cell surface antibodies to obtain integrin beta-7^+^ CD31^−^ CD45^−^ CD29^+^ CD56^+^ cells [3,11] and the pre-plating isolation method using a difference in cell adhesion rates [14,15]. However, for cell-cultured meat, which requires a large number of cells to be productive, these two methods are inadequate. The method using a specific antibody exhibited a low recovery rate, high cost, and safety concerns because the isolation tube is blocked by muscle fibers developed in adult cattle; however, the pre-plating isolation method required effort and time and had a low recovery rate. According to Furuhashi et al. [19], the skeletal muscle tissue comprises more than 80% muscle cells and approximately 12% fibroblasts and vascular endothelial cells. The expression patterns of muscle transcription factors were analyzed according to isolation methods, such as the simple method to remove fibroblasts and the sorting method using specific antibodies. Using the simple method to remove fibroblasts from skeletal muscle cell suspension improved their differentiation ability under DM conditions. This method can make a significant contribution in terms of improving the productivity of cell agriculture without the loss of cells.

In general, SKMCs are cultured in DMEM with ≥20% serum, which is a barrier to the mass production of cultured meat [1,2]. Using FBS has difficulties in terms of the danger of contracting zoonosis and animal welfare violations associated with animal cruelty. Moreover, it is costly and unsustainable. Recently, a B8 culture medium developed for induced pluripotent stem cell (IPS) cell culture [5], B9 culture medium improved from B8 (B8 supplemented with rat albumin), and XENO-free culture medium [4] have improved upon these disadvantages; however, they demonstrate a decrease in cell growth rate under long-term culture conditions [1,2]. In this study, physiological hypoxia (2–9%), similar to the oxygen pressure in skeletal muscle in vivo, was used to reduce the amount of serum used for cell growth [7,20]. Hypoxia, along with 5% serum, resulted in a cell number similar to that under normoxia in 20% serum, confirming that hypoxia can reduce serum requirement (~4-fold). In addition, at the same serum concentrations, 5% hypoxia increased cell number by approximately 2-fold of that under normoxia. Hypoxia aids in the reduction of FBS levels required in GM for SKMC development. This result is supported by the close interaction effect of hypoxia levels and serum concentration.

For a comprehensive analysis, the effect of serum contained in GM on MRFs was analyzed at the molecular level; 20% serum blocked the differentiation pathway by inhibiting *MRF* (*MYF5*/*MYOD1*/*MYOG*) and *MYH1* genes. In contrast, additional physiological hypoxia at low serum concentrations (≤10%) enhanced the upregulation of genes encoding MRFs and *MYH*. Therefore, this study revealed that the MRF pathway relies more on serum than on oxygen for SKMC proliferation. In addition, this study analyzed the genes involved in cell lifespan and ROS production in vitro. Elevated serum levels (20%) lowered the expression of *TERT*, which is involved in the synthesis of telomerase. By repetitive cell proliferation, p21 stability in the TP53-CDKN1A (P21) pathway might compensate for the extra physiological hypoxia at high serum concentrations. Elevated serum levels suppressed ROS by reducing *NOX2* expression and activating superoxide dismutase 2/3. These results are supported by current literature that indicates that albumin, a major component of serum, has antioxidant effects [1,21,22]. Physiological hypoxia at elevated serum levels had a synergistic effect on inhibiting potential cell differentiation by downregulating *p21*/*SODs* and maintaining *MSTN* expression in vitro.

The primary trigger for myoblast differentiation is the induction of a starving state by a reduction in serum levels. Myoblast differentiation is also influenced by hypoxia and the composition of the in vitro DM [10]. Serum starvation (≤0.5% FBS) successfully induced multinucleated myotube cells in SRCs but demonstrated considerable cell death, thereby reducing the total number of nuclei. The findings indicate that multinucleated myotube cell purity was increasing; however, the total cell number related to cell productivity was decreasing. There is little doubt that physiological hypoxia, rather than normoxia, is beneficial for maintaining and proliferating myoblasts, although there are contradictory studies on the influence of physiological hypoxia on myoblast/muscle satellite cell differentiation [10,12,23]. Regarding myoblast differentiation, the negative effect of physiological hypoxia is mostly observed in C2C12 cells, an immortalized mouse myoblast cell line; however, its positive effects were primarily confirmed in primary myoblasts from animals [10,11,12,13]. Physiological hypoxia and the culture medium of the composition are factors that have a major influence on the in vitro differentiation of myoblasts [7,10,24]. This study found that the composition of the DM is a factor that is considerably more critical than physiological hypoxia which is facilitated by altering the expression of genes involved in MRFs and *MYH1* (upregulation by 15.4–63.7-fold)/*MSTN* in DMEM/F12 DM, including high concentrations of glucose, amino acids, and vitamins. Under optimal culture medium conditions for SKMC differentiation, physiological hypoxia was a positive factor for *MYH* but a negative factor for *MSTN*.

In conclusion, serum levels were found to be the key regulator for the optimum proliferation and differentiation of bovine SKMCs, and physiological hypoxia in serum was a synergistic component. In addition, it was confirmed that a DM rich in amino acids and vitamins was a more important factor in the differentiation of bovine SKMCs than physiological hypoxia under low serum conditions (Figure 10). A cellular agriculture application using bovine SKMCs should be considered owing to its large-scale production, low production costs, and animal welfare. Thus, for price competitiveness and safety, we will pursue research to produce serum substitutes derived from plants or yeast and to convert major components of the culture from laboratory grade to food grade.

## Figures and Tables

**Figure 1 foods-12-01384-f001:**
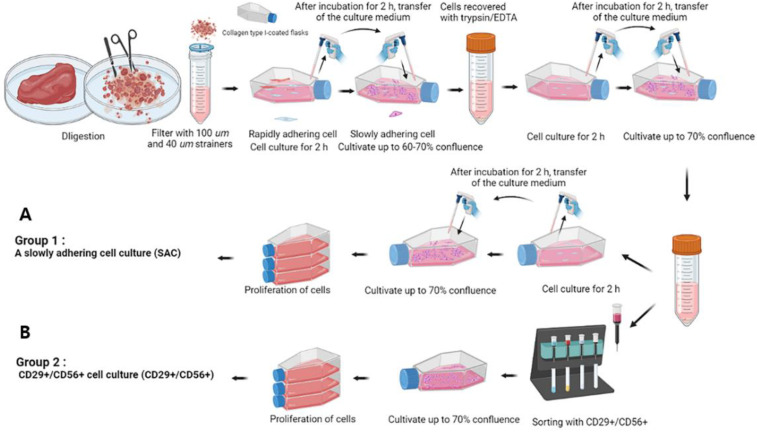
Strategies for the recovery of skeletal muscle tissue-derived myogenic cells (SKMCs) from Hanwoo Korean native cattle. Skeletal muscle tissue was physically chopped, and after 0.1% collagenase treatment for 30–60 min at 37 °C, they were passed through 100- and 40-µm filters sequentially to remove less degraded tissues. (**A**) Group 1 was named the slowly adhering cell (SAC) culture method. SACs were recovered after three attempts of negative selection of fast-adhering cells, and they were either cryopreserved or proliferated for further experiments. (**B**) Group 2 was named the CD29^+^/CD56^+^ cell culture method. This method was performed with a positive selection of CD29^+^/CD56^+^ cells after two trials of negative selection of fast-adhering cells. The schematic of cell separation was created using the free version provided by BioRender.com (accessed on 11 January 2023).

**Figure 2 foods-12-01384-f002:**
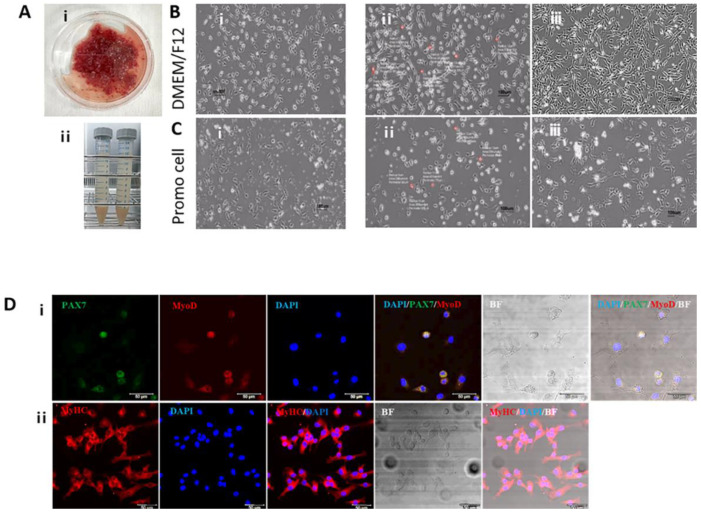
Morphology of SKMCs in growth medium (GM). (**A-ⅰ**), Chopped skeletal muscle tissue recovered from Hanwoo; (**A-ⅱ**), recovered SKMCs. The cells were cultured with DMEM/F12 with 20% FBS (**B**) or PromoCell with 5% FBS (**C**). Cells that attached quickly were cultured for 5 days in both groups (**B-i**,**C-i**). Slowly attaching cells were cultured for 5 days in both groups (**B-ii**,**C-ii**), and additionally for 2 days in both groups (**B-iii**,**C-iii**). Scale bars = 100 μm. (**D**) Immunofluorescence staining was performed on the isolated cells with specific PAX7 (FITC, green)/MYOD (Alex Fluor 594, red) (**ⅰ**) (satellite cell transcription factors) and MyHc (Alex Fluor 594, red) (**ⅱ**) antibodies. BIF is brightfield microscopy and DAPI (blue) is used for cell counting. The figures show merged images for different combinations, such as PAX7/MyoD/DAPI, PAX7/MyoD/DAPI/BIF, MyHC/DAPI, and MyHC/DAPI/BIF. Scale bars = 50 μm.

**Figure 3 foods-12-01384-f003:**
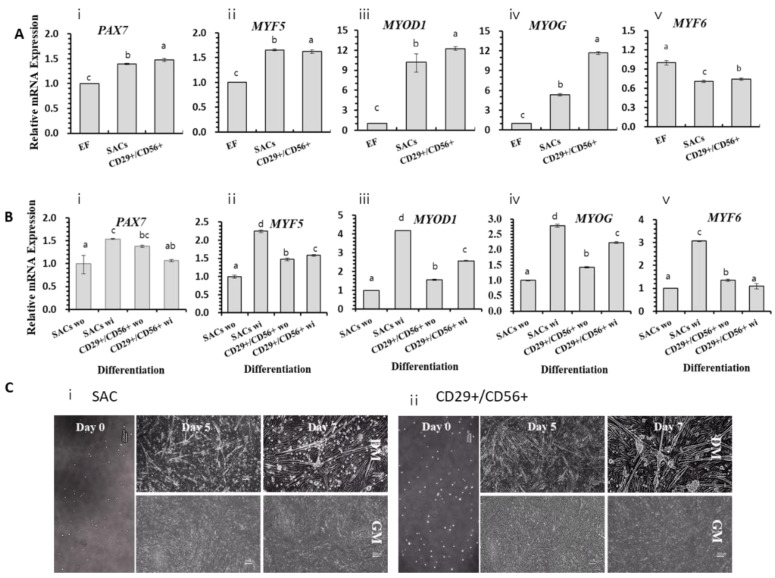
Comparison of expression levels of myogenic regulatory factors (MRFs) on in vitro proliferation and differentiation of SKMCs based on the method of cell isolation. (**A**) Cells were divided into SAC and CD29^+^/CD56^+^ groups, cultured in GM for 7 days, and genes encoding MRFs were analyzed by real-time PCR. (**B**) Cells that reached 90% confluence were cultured in GM (without, wo) or DM (with, wi) for 7 days, and genes encoding MRFs were analyzed by real-time PCR. (**i**–**v**) in (**A**,**B**) represent the PAX7 (**i**), MYF5 (**ii**), MYOD1 (**iii**), MYOG (**iv**), and MYF6 (**v**) genes, respectively. (**C**) Morphological changes of cells cultured with GM or DM were shown over time. (**C-i**,**C-ii**) represent the SAC and CD29^+^/CD56^+^ groups, respectively. EF inbdicates ear fibroblasts isolated from Hanwoo. All experiments were performed using five replicates. Data are expressed as relative quantification (RQ), and error bars indicate minimum and maximum values of RQ. ^a–d^ *p* < 0.05. GM and DM indicate growth medium (DMEM/F12-based medium with 20% FBS) and differentiation medium (DM; PromoCell-based DM), respectively. Scale bars = 50 µm.

**Figure 4 foods-12-01384-f004:**
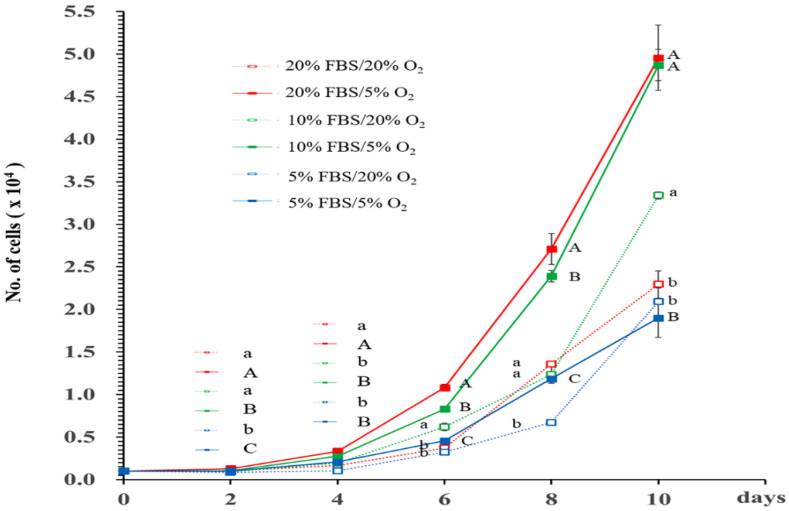
Effects of FBS and oxygen as major influencing factors on cell number increase during in vitro growth of SKMCs. SKMCs were cultured in GM with varying oxygen (5, 20%) and FBS (5, 10, 20%) concentrations for ten days, and the number of cells was counted every two days. All analyses were performed up to day 10 in two-day intervals using five replicates. Data are expressed as mean ± standard error of mean (SEM). ^A–C^ *p* < 0.05. ^a,b^ *p* < 0.05.

**Figure 5 foods-12-01384-f005:**
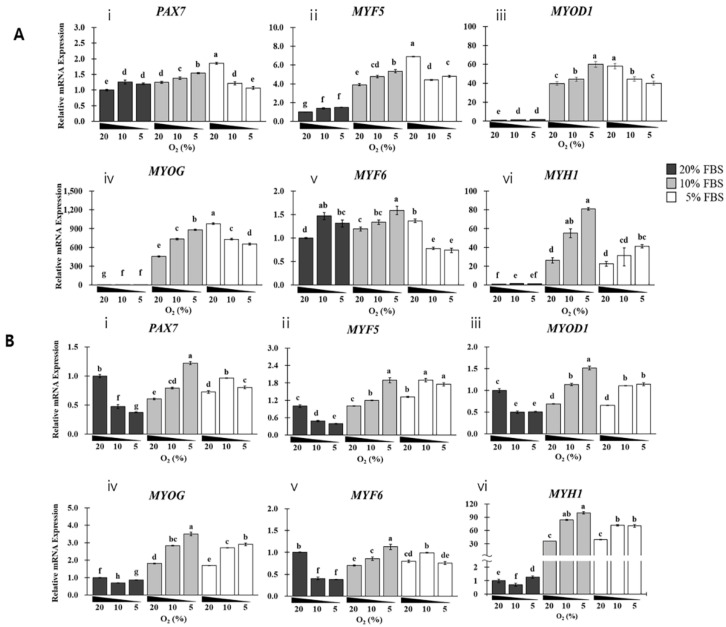
Effect of FBS and oxygen on the expression of the genes encoding MRFs during in vitro growth of SKMCs for 14 days. SKMCs were cultured in GM with FBS (5, 10, and 20%) under additional physiological hypoxic conditions (5, 10, control 20%) for 14 days and analyzed on days 7 (**A**) and 14 (**B**). All analyses were performed in five replicates. Labels i–vi on (**A**,**B**) correspond to the expression of *PAX7* (**i**), *MYF5* (**ii**), *MYOD1* (**iii**), *MYOG* (**iv**), *MYF6* (**v**), and *MYH1* (**vi**), respectively. The black, gray, and white bars on the histogram indicate that 20%, 10%, and 5% of FBS were added to GM, respectively. Data are expressed as RQ, and error bars indicate minimum and maximum values of RQ. ^a–g^ *p* < 0.05.

**Figure 6 foods-12-01384-f006:**
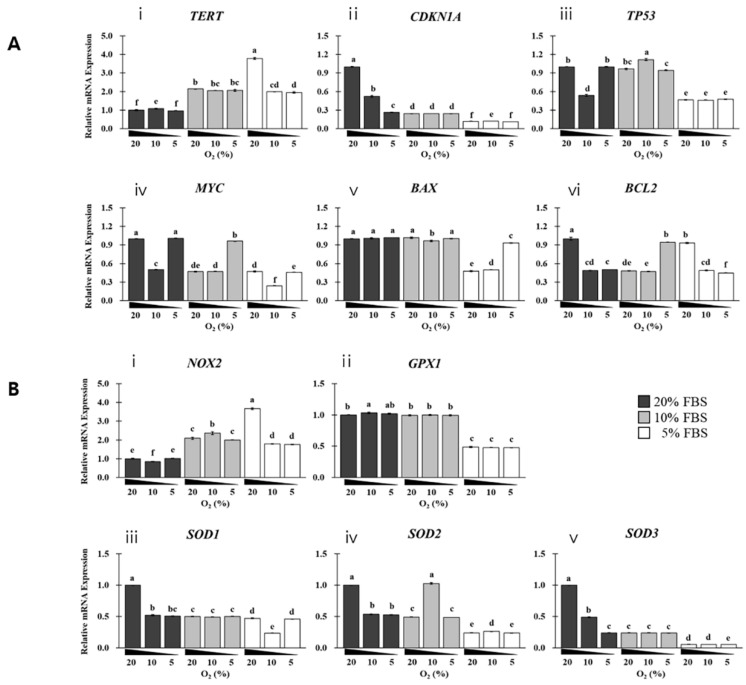
Effect of FBS and oxygen on the expression of genes involved in proliferation, apoptosis, or reactive oxygen species (ROS) generation during in vitro growth of SKMCs. SKMCs were cultured under additional physiological hypoxia induction (5%, open bars; 10%, gray bars; control 20%, black bars] in GM with FBS (5, 10, and 20%) for 14 days. The expression of each gene was analyzed by real-time PCR. (**ⅰ**–**ⅵ**) of (**A**) indicate *TERT*, *CDKN1A*, *TP53*, *MYC*, *BAX*, and *BCL2*, respectively. (**ⅰ**–**ⅴ**) of (**B**) indicate *NOX2*, *GPX1*, *SOD1*, *SOD2*, and *SOD3*, respectively. All analyses were performed with five replicates. The black, gray, and white bars on the histogram indicate that 20%, 10%, and 5% of FBS were added to GM, respectively. Data are expressed as RQ, and error bars indicate minimum and maximum values of RQ. ^a–f^ *p* < 0.05.

**Figure 7 foods-12-01384-f007:**
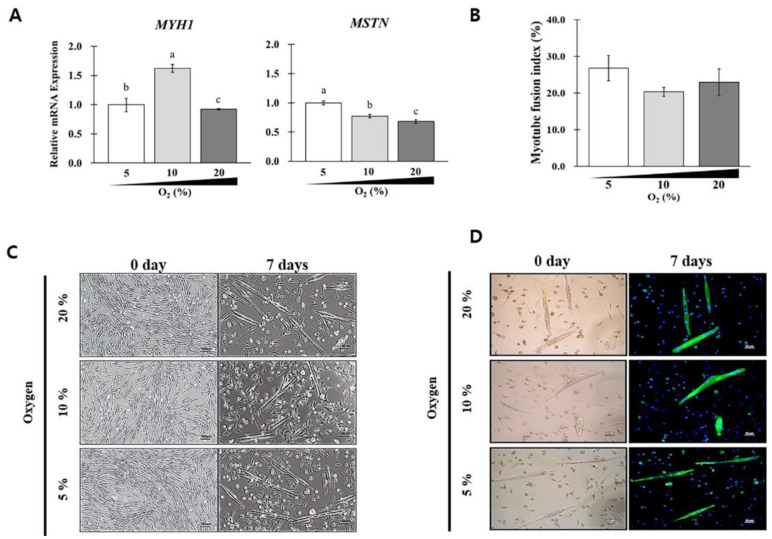
Evaluation of in vitro differentiation capacity of SKMCs cultured in serum-free PromoCell DM. SKMCs were cultured in serum-free PromoCell DM under different oxygen concentration conditions (5, 10, and 20%) for 7 days. (**A**) Expression of *MYH1* and *MSTN*; (**B**) myotube fusion index (MFI); (**C**) cell morphology; (**D**) multinucleated myotube cells by immunocytochemistry staining. Dark gray, gray, and white bars in the histogram represent 20%, 10%, and 5% oxygen concentrations, respectively. The cells were treated with anti-myosin (Skeletal, Fast) antibody (dilution, 1:400; Sigma; MyHC) and followed by the reaction of goat anti-mouse IgG (H+L)-Alexa Fluor 488 (dilution, 1:200; Invitrogen) (green). The nuclei were counterstained with DAPI for 30 min. The samples were observed under a fluorescence microscope. MFI was calculated by dividing the number of nuclei within the myotubes (with three or more nuclei) in MYHC-positive myotubes by a total number of nuclei × 100. This experiment was repeated thrice. Data are expressed as mean ± SEM. ^a–c^ *p* < 0.05.

**Figure 8 foods-12-01384-f008:**
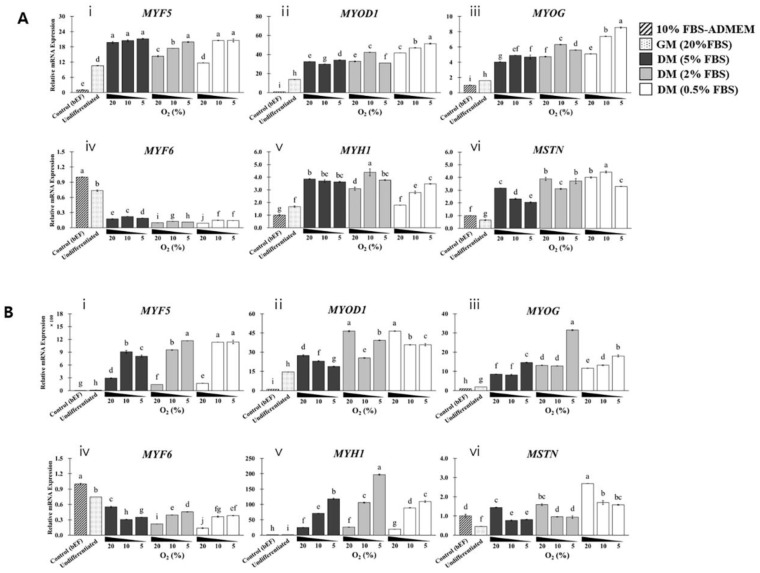
Comparison of in vitro differentiation capacity of SKMCs according to oxygen, FBS, and DM. All SKMCs were differentiated with F10 basal medium (**A**) or DMEM/F12 basal medium (**B**) for 7 days. FBS concentrations in DM were 0.5% (open bars), 2% (gray bars), or 5% (black bars). Oxygen concentrations were 20, 10, or 5%. The control group was fibroblasts recovered from the same subject; they were cultured in advanced DMEM with 10% FBS (dotted bars). “Undifferentiated” indicates SKMCs cultured in GM with 20% FBS for 7 days (oblique line bars). Labels (**i**–**vi**)on A and B indicate *MYF5*, *MYOD1, MYOG, MYF6, MYH1*, and *MSTN*, respectively. All analyses were performed with five replicates. Data are expressed as RQ, and error bars indicate minimum and maximum values of RQ. ^a–g^ *p* < 0.05.

**Figure 9 foods-12-01384-f009:**
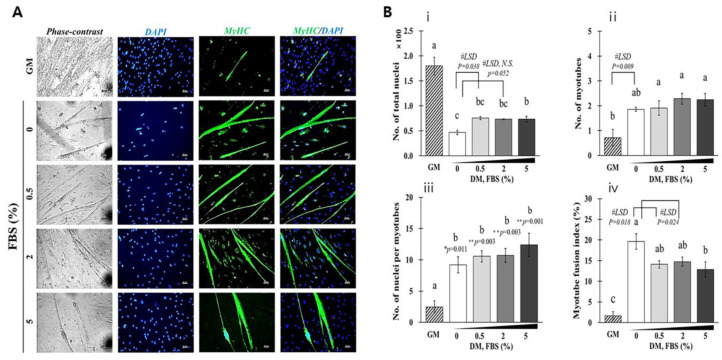
MFI in SKMCs differentiated with DMEM/F12 DM. All SKMCs were differentiated with DMEM/F12 basal DM with 0, 0.5, 3, or 5% FBS for 7 days in 20% oxygen and fixed. (**A**) The cells were treated with anti-myosin (Skeletal, Fast) antibody (dilution, 1:400; Sigma; MyHC), followed by a reaction with goat anti-mouse IgG (H+L)-Alexa Fluor 488 (dilution, 1:200; Invitrogen) (green). The nuclei were counterstained with DAPI for 30 min. The samples were observed under a fluorescence microscope. MFI (**B**-**iv**) was calculated by dividing the number of nuclei within the myotubes (with three or more nuclei) (**B**-**iii**) in MYHC-positive myotubes (**B**-**ii**) by the total number of nuclei (**B**-**i**) ×100. This experiment was repeated thrice. The white, light gray, dark gray, and black bars in the histogram indicate that DM contained 0%, 0.5%, 2%, and 5% FBS, respectively, while the hatched bar represents the GM medium. Data are expressed as mean ± SEM. Post hoc analysis (Tukey’s test) with one-way ANOVA (^a–c^ *p* < 0.05) was used to analyze the data. Some of the data were significant based on the posthoc LSD test (#). * and ** indicate significance levels at *p* < 0.05 and *p* < 0.005, respectively.

**Figure 10 foods-12-01384-f010:**
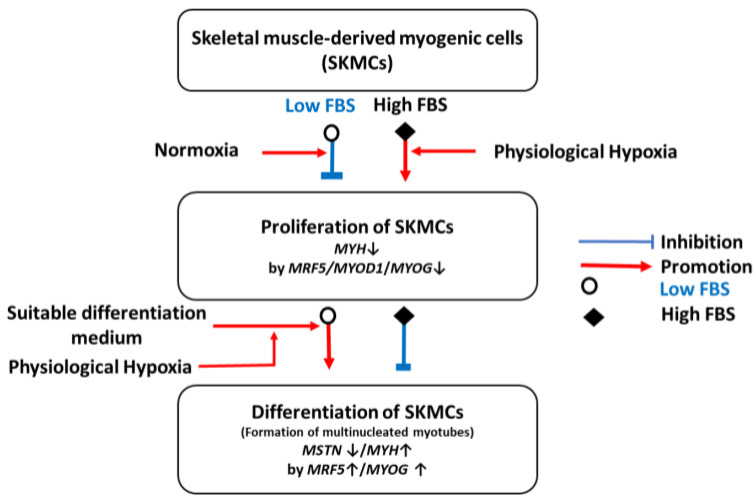
Relationship between hypoxia, serum, and DM in vitro growth and differentiation of SKMCs obtained from Hanwoo Korean native bull.

## Data Availability

All data generated or analyzed during this study are included in this published article and its Appendix A.

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
