# Peer review of "Effect of Serum and Oxygen on the In Vitro Culture of Hanwoo Korean Native Cattle-Derived Skeletal Myogenic Cells Used in Cellular Agriculture"

_foods, 2023, doi:10.3390/foods12071384_

Round 1
Reviewer 1 Report
The report by Ock et al, describes a study in bovine primary myoblasts on the effect of serum concentration and hypoxia on proliferation and differentiation. it follows some pre-existent studies with -as referenced- conflicting results, especially regarding the influence of hypoxia. The authors use primary bovine cells and the main criticism is that only donor has been used, ignoring donor-to-donor variation. The major outcome is that hypoxia reduces the need for serum and that differentiation is medium dependent. The effect of hypoxia on differentiation is ambiguous.
minor comments
- grammar needs to be checked
- SMC typically refers to smooth muscle cells and not skeletal muscle cells
- line 83. anti-anti is only later explained-
- methods: coating of flasks?
- figure captions: 5 repetitions refer to what? cell culture wells or measurements?
- line 116 concentration of EGF correct?
- fusion index. Was counting of myotubes done manually? If so, were observers blinded to condition?
- figure 2: are there two different legends?
- figure 4B: redundant information; same as figure A
- figure 5 legend. a-d p<0.05, how about the groups e through g?
- results from figure 6 and 7. How are the results interpreted in view of different influences on cell proliferation and differentiation. If not related, these additional measurements are irrelevant for the conclusion
- Figs 8d, 10b. Very few cells participate in myotube fusion. Are these results relevant for a cultured meat process?
Discussion: the discussion is hard to read and does not lead to a conclusion.
line 495: are these the only differences between Promocell and DMEM/F12?
lines 498-502: this final paragraph on albumin seems unrelated to the rest of the manuscript
Reviewer 2 Report
General comments:
This is an important paper with relevant information for the continued research that is required for cellular agriculture to flourish. As suggestions for improvement, some point can be raised. More information is required in the section Material and Methods. The first part of the Results section is confusing, requiring attention to the text and to the figures as well, as per details below. In addition, authors may consider two figures to show: 1. the genes studied together with their main effects on SMC proliferation and differentiation, as well as their response as FBS and oxygen percentages increase or decrease and the interaction FBS x oxygen; 2. the main conclusions in a visual format; this may significantly improve paper readability. Some of the graphs could be switched to the supplementary material, to make room for these figures. Please check more detailed comments below.
Specific comments:
L14-15 Please check written text, as the verb ‘to examine’ is repeated.
L18 and throughout the text: do not start phrases with acronyms.
L43: Phrase is not clear, please improve.
L74: Do not call the animal “a cow” if it is a male.
L75: Please be more technical and specific in describing the muscle from which the cells originated - it is not enough to describe the hamstring, as this is the popular name of a group of muscles.
L193: Is this a different figure 2 from the one on page 4? Where is this figure? Please check figure numbering and presentation.
L219-228: Please improve the title of Figure 3, more explicitly stating the details of part A and B as well as other details relevant to the understanding of each part of the figure. Please name the y axis of parts A and B. Also, please mention the use of ear fibroblast in the methodology, in terms of motivation for its inclusion, of sample collection method (or origin if not collected by you) and of lab maintenance.
L230-231: Please provide all required details in the Material and Methods section and do not repeat methodology here.
L235-236: Please check grammar.
L240-241: Please make figure title more informative, by adding details such as representation of statistical significance in the figure. It is not clear what the difference between A and B parts of this figure is. Please either make this explicit or remove one part, leaving only one graphical representation - probably B, which makes more sense, as connecting the dots with a line usually refers to events across time, and also makes differences more evident by using the same y axis scale.
L242-247: Bring this paragraph to the previous one, rewrite for text fluency.
L446: The main reason why FBS violates animal welfare is because it involves animal cruelty - please insert this in addition to cost and sustainability issues.
Round 2
Reviewer 1 Report
The manuscript has improved and errors have been corrected. It is good to know that results were based on 3 donors and typically 3 wells per cell culture. It is in my view not acceptable to list the results of only one donor and state that the others were similar. You really should report a compound result from 3 donors. For reporting number of culture wells, legends should indicate that, or at the very least it should be mentioned in the methods (e.g. PCR section); this is more relevant than 5 technical replicates.
minor remarks
- concentration of growth factors in materials and methods is still unclear. "10 ng/mL 129 bFGF/hepatocyte growth factor (HGF)/insulin-like growth factor 1 (IGF1)/EGF" is this 10ng/ml for each GF or 10ng/ml for the mix?
- fusion index assessment: it is not sufficient to refer to a reference, please state explicityly that analysis was done blinded to treatment group, if that was indeed the case.
- figure 8 A/B Legend. A how many days?, B what medium?
Author Response
Dear Reviewer
We wrote the corrections in detail in the attached file below and included references for better understanding.
Sincerely
